# Validating an Instrument for Direct Patient Reporting of Distress and Chemotherapy-Related Toxicity among South African Cancer Patients

**DOI:** 10.3390/cancers14010095

**Published:** 2021-12-24

**Authors:** Charmaine L. Blanchard, Keletso Mmoledi, Michael H. Antoni, Georgia Demetriou, Maureen Joffe, Gilberto Lopes, Paul Ruff, Daniel S. O’Neil

**Affiliations:** 1Centre for Palliative Care, Department of Medicine, Faculty of Health Sciences, University of the Witwatersrand, Johannesburg 2193, South Africa; charmaine.blanchard@wits.ac.za; 2Non-Communicable Disease Research Division, Wits Health Consortium (PTY) Ltd., Johannesburg 2193, South Africa; keletso.mmoledi@wits.ac.za (K.M.); mjoffe@witshealth.co.za (M.J.); Paul.Ruff@wits.ac.za (P.R.); 3Sylvester Comprehensive Cancer Center, University of Miami Health System, Miami, FL 33136, USA; mantoni@miami.edu (M.H.A.); glopes@med.miami.edu (G.L.); 4Department of Psychology, University of Miami, Miami, FL 33136, USA; 5Division of Medical Oncology, Department of Medicine, Faculty of Health Sciences, University of the Witwatersrand, Johannesburg 2193, South Africa; georgia.demetriou@wits.ac.za; 6MRC Developmental Pathways to Health Research Unit, Department of Pediatrics, Faculty of Health Sciences, University of Witwatersrand, Johannesburg 2193, South Africa

**Keywords:** patient reported outcomes, chemotherapy toxicity, distress, quality of life, South Africa

## Abstract

**Simple Summary:**

In the USA and Europe, questionnaires that allow patients to directly report chemotherapy side effects can improve their quality of life and clinical outcomes. No similar tools exist for Africa, so we aimed to design and validate a paper-based tool for use in oncology clinics in South Africa: The Patient Reported Symptoms-South Africa (PRS-SA) survey. The PRS-SA included questions on overall feelings of distress and severity of 11 common chemotherapy side effects. By comparing responses to the PRS-SA to responses to other quality of life questionnaires and to patients’ performance status, we found the PRS-SA to be valid and responsive for measuring all included symptoms. Compared to a standard instrument for measuring depression and anxiety, the PRS-SA’s distress thermometer had 88% sensitivity and 55% specificity for identifying distress. The PRS-SA may be a useful tool for efficiently assessing distress and chemotherapy symptoms in South Africa’s overburdened public oncology clinics.

**Abstract:**

Patient-reported outcome measures (PROM) for monitoring treatment toxicity improve quality of life (QoL) and clinical outcomes. However, no such PROMs exist for sub-Saharan African cancer patients. We aimed to validate the Patient Reported Symptoms-South Africa (PRS-SA) survey, a novel PROM for measuring distress and chemotherapy-related symptoms in South African cancer patients. We enrolled patients at the oncology clinic at Charlotte Maxeke Hospital, Johannesburg. At three separate visits, participants simultaneously completed the PRS-SA survey and several previously validated questionnaires. We constructed a receiver operator characteristics curve for distress levels predicting a Hospital Anxiety and Depression Scale (HADS) score ≥15. We evaluated construct validity for symptom items by comparing severity to the EORTC Core Quality of Life Questionnaire (QLQ-C30) summary score (Pearson correlation tests) and ECOG performance status (Mann–Whitney U tests). We assessed symptom item responsiveness by comparing change in severity to change in QLQ-C30 summary score and comparing standardized mean scores with negative, no, or positive change on the Global Impression of Change (GIC) questionnaire (Jockheere–Terpstra trend test). Overall, 196 participants with solid tumors completed instruments. A distress score of 4 had 82% sensitivity and 55% specificity for clinical depression/anxiety. All symptom items showed construct validity by association with either QLQ-C30 score or performance status (highest *p* = 0.03). All but cough showed responsiveness to change in QLQ-C30 score (highest *p* = 0.045). In South African cancer patients, the PRS-SA’s stress scale behaves similarly to the distress thermometer in other populations, and the symptom items demonstrated construct validity and responsiveness. Of note, 46% and 74% of participants who completed the PRS-SA in English or isiZulu, respectively, required assistance reading half or more of the instrument.

## 1. Introduction

Use of patient-reported outcome measures (PROMs) has been expanding into routine clinical oncology practice, particularly for monitoring cancer- and chemotherapy-related symptoms. Various academic cancer centers have developed electronic, web-based programs that allow patients to directly record the types and severity of symptoms they are experiencing while undergoing therapy [1,2,3]. When compared to symptoms documented by physicians alone, patient-reporting consistently captures a greater overall symptom burden, suggesting that usual care underestimates the true extent of cancer- and therapy-related symptoms [4,5]. Patient–provider communication is improved, as physicians are able to devote more time to discussing the management of patient-reported symptoms [4,6]. Patient satisfaction with such systems is consistently high [3,6].

PROMs can also have clinical impact [6]. At Memorial Sloan Kettering, advanced cancer patients randomized to a web-based symptom reporting system experienced higher quality of life (QoL), fewer emergency room visits, fewer hospitalizations, longer periods of receiving recommended chemotherapy and even longer overall survival [7,8].

In addition to physical symptoms, mental and emotional distress is prevalent among cancer patients, with broad meta-analyses suggesting up to 40% of cancer patients may simultaneously experience depression, anxiety or adjustment disorder [9]. Heightened distress is associated with reduced overall survival for a variety of cancers [10]. Screening for distress, particularly when followed by referral to psychological interventions, can reduce subsequent distress levels and may even improve short-term survival [11,12]. The National Comprehensive Cancer Network (NCCN) recommend that patients be screened for distress at every medical visit, or, at a minimum, at their initial visit and appropriate intervals [13].

As far as we know of, there are no reports from sub-Saharan Africa (SSA) of systematic PROM use in cancer care. Frequently, in the resource-constrained healthcare settings of SSA, patient volumes are extremely high, and time spent on each patient-provider interaction is limited. However, given the potential of PROMs to improve patient–provider communication and identify patients appropriate for additional supportive interventions, they may be an ideal, low-tech tool for strengthening those interactions and improving patient QoL and outcomes.

We designed a paper-based PROM for monitoring physical symptoms of chemotherapy toxicity and patient distress levels called the Patient-Reported Symptoms-South Africa (PRS-SA) survey. We aimed to validate that instrument for use in South Africans undergoing treatment for solid tumors.

## 2. Materials and Methods

### 2.1. Patient-Reported Symptoms-South Africa Instrument Design

We designed our novel instrument by adapting PROM tools that have been successful in other, high-income settings. The authors prepared an initial draft instrument containing the elements described below. An informal group discussion was held with a convenience sample of South African cancer and palliative care physicians, nurses, and patient navigators to review the draft and elicit feedback on completeness and relevance of the content and appropriateness of the language. That feedback was then incorporated to create a finalized version (Appendix A).

To capture overall distress, we used a version of the Distress Thermometer (DT) instrument. The classic DT is a visual analogue scale labelled with scores from 0 (‘no distress’) to 10 (’extreme distress’) and was developed by Roth A.J. et al. in 1996 as a simple screening tool for identifying cancer patients needing further review for depressive or anxiety symptoms [14]. The version included in the NCCN Clinical Practice Guidelines in Oncology: Distress Management report also asks patients to select items from a list of specific problems potentially contributing to distress (e.g., practical, family, emotional, spiritual, and physical) [13]. The DT has now been translated and validated for use in many European, Asian and American populations [15].

Our group of providers believed that the term “suffering” would be best understood by the local population to represent the combination of physical, emotional, and spiritual discomfort characterized in the United States as distress, so we labelled our DT as ranging from “No Suffering” to “Severe Suffering.” The list of practical problems was reduced and consolidated as well.

To capture the severity of common chemotherapy-related symptoms (CRS), we modeled the approach of Basch et al. [16]. First, we selected eleven frequent CRS: pain, fatigue, fever, dyspnea, cough, oral mucositis, nausea/anorexia, vomiting, diarrhea, constipation and peripheral neuropathy. We then adapted the National Cancer Institute’s Common Terminology for Adverse Events, version 5.0′s graded severity descriptions into patient-friendly language that also reflected local healthcare practice [17]. After the language was reviewed and modified with our provider group, the severity descriptions were laid out in grid with instructions to select the severity of each symptom experienced since the most recent dose of chemotherapy or over the prior seven days if chemotherapy had not yet been started.

The final English draft underwent forward–backward translation into isiZulu, the most commonly spoken local language in the Gauteng province. Both English and isiZulu versions were then piloted with five patients each to gauge understanding of the content; no changes were made based on this last step.

### 2.2. Additional Study Instruments

Similar to the approach taken by the National Institute of Health PRO-CTCAE Study Group, we evaluated the construct validity and responsiveness of each CRS item by comparing participant responses to previously validated “anchor” instruments and to pre-defined clinical sub-groups expected to have differing symptom burden [18]. Our anchor instruments included the European Organisation for Research and Treatment of Cancer Core Quality of Life Questionnaire, version 3.0 (QLQ-C30) and a single-item version of the Global Impression of Change (GIC) questionnaire [19,20]. Our clinical grouping was limited to Eastern Cooperative Oncology Group performance status (ECOG PS), as assessed by study staff [21].

The QLQ-C30 is a cancer-specific quality of life instrument that measures patient experience within five functional categories, eight symptom categories and global health status; the instrument specifically asks about experience of the past week. A single health-related QoL summary score can be calculated by linearly transforming and then averaging the individual scores from each functional and symptom category, excluding financial difficulties [22,23]. The QLQ-C30 has an official isiZulu language version and has been previously administered in South Africa [24,25].

The GIC simply asks patients to describe their “overall status” since the last time they were surveyed according to 7 response choices ranging from “Very Much Worse” to “Very Much Improved.” We prepared an isiZulu version of the GIC, also using the forward–backward translation method.

Validation of the modified DT focused on establishing a threshold value with balanced sensitivity and specificity for detecting clinically meaningful depression and anxiety. As with the original and most of the subsequent validation studies of the DT, we used the Hospital Anxiety and Depression Scale (HADS) as a comparator instrument [15,26]. The HADS is a 14-item survey with 7 questions each evaluating depression and anxiety related symptoms experienced over the past week [27]. Each item is rated on a scale of 0 to 3, with a total score of ≥15 signaling significant distress. The HADS has previously been validated in South Africa [28].

### 2.3. Setting and Participants

Taken in total, South Africa is an upper middle-income country, but the stark degree of income inequality means that 56% of the population live below the national poverty line [29]. South Africa has 11 official languages and at least 25 commonly spoken languages [30]. The most common first language is isiZulu, spoken by 23% at home and understood by over 50%. English is the 4th most common first language in the country, spoken by 10% at home, and the 2nd most common in Johannesburg’s Gauteng Province. English is also the standard language used in educational, governmental and business activities. As of 2019, 88% of South Africans over 15 years old had completed Grade 7, signifying functional literacy [31].

Low-cost cancer care is available to uninsured and underinsured patients at public tertiary hospitals. For this study, we enrolled participants from the medical oncology clinic of one such hospital: Charlotte Maxeke Johannesburg Academic Hospital (CMJAH). The hospital is an affiliated teaching hospital of the Faculty of Health Sciences at the University of Witwatersrand. The clinic provides chemotherapy and other systemic cancer therapies to most public patients in Johannesburg, including those who receive their surgical care at Chris Hani Baragwanath Academic Hospital, an enormous public hospital serving Johannesburg’s majority Black neighborhood of Soweto. Patient–provider ratios are high in South Africa’s public settings [32]. Long wait times at specialist clinics often result in very brief face-to-face patient interactions.

Participants were eligible if they were ≥18 years of age, had a histologically confirmed diagnosis of a non-hematologic cancer, were either currently receiving chemotherapy at CMJAH’s oncology clinic or scheduled to initiate chemotherapy within 14 days of enrollment, were not receiving their last planned dose of chemotherapy on the day of enrollment and were able to communicate in either English or isiZulu. Patients with both metastatic and non-metastatic disease were allowed to enroll.

### 2.4. Study Procedures

Potentially eligible clinic patients were identified by study staff and recruited on the day of their normally scheduled clinic visit. No randomization was performed for this single arm study and consenting patients were enrolled consecutively. Patients were approached in the waiting area and, if they expressed interest in participation, were taken to a private room for the consenting process and the administration of study questionnaires.

On the day of enrollment, consenting and eligible participants completed, in order, the PRS-SA, QLQ-C30 and HADS questionnaires. The study interviewer collected self-reported data on demographics, literacy, and basic clinical information. Participants were then followed for up to two subsequent clinic visits. At those visits, they completed the PRS-SA, QLQ-C30 and GIC questionnaires (Figure 1). Interviewer-assessed ECOG PS was also captured at each study visit. As study procedures occurred at participants’ regular clinic visits on the day of chemotherapy infusion, they typically occurred every 3 weeks.

Study data were collected and managed using REDCap electronic data capture tools hosted at the University of Witwatersrand [33]. To be consistent with the intended clinical use, the PRS-SA was completed on paper and responses were subsequently entered into the study database. Participants were given the opportunity to read and complete the PRS-SA unassisted but were also instructed that study interviewers could read some or all the instrument to them if requested. Following each visit, the interviewer documented the degree of assistance provided with reading the PRS-SA.

### 2.5. Statistical Analysis

Patient characteristics, literacy skills and reported levels of distress and physical symptoms were described using simple counts and percentages.

Construct validity for each symptom item was assessed separately by comparing individuals’ reported severity for each item on the PRS-SA to the QLQ-C30 summary score and ECOG PS (PS 0–1 vs. 2–4) measured at the same visit. Pearson correlations with 95% confidence intervals were calculated for the comparison to QLQ-C30, and the Mann–Whitney U tests were used to evaluate association with performance status. In addition to testing for significant association, we determined the strength of association between each PRS-SA item and the anchor item. Applying the same standard as the PRO-CTCAE Study Group, for the QLQ-C30 and ECOG PS comparisons we, respectively defined r values of 0.1, 0.3 and 0.5 and Cohen d effect sizes of 0.2, 0.5 and 0.8 as small, medium, and large effects. If a PRS-SA symptom item showed a significant association with at least a small effect size for either anchor item, we considered it to have shown construct validity. As sensitivity analyses, we also calculated Pearson correlations between all PRS-SA symptom item scores and the individual QLQ-C30 functional sub-scale (i.e., physical, role, emotional, cognitive and social functioning) and any relevant symptom sub-scale scores.

Responsiveness for each symptom item was assessed in two ways. First, for each item, we calculated the change in reported severity on the PRS-SA and change in QLQ-C30 between pairs of consecutive visits. Pearson correlations were calculated to evaluate for association between the changes in those two values. In addition, individual visits were sorted into three subgroups based on response to the GIC questionnaire (global improvement, no change or global decline). Within each group, for each individual symptom item, we calculated the standardize mean change in severity from the prior visit. Those standardized means were compared across GIC subgroups using a 1-sided Jockheere–Terpstra test for trend. We considered items responsive if we found significant correlation for change in severity and QLQ-C30 score and found the Jockheere–Terpstra test to be significant (Figure 1). All statistical tests were two-sided, unless otherwise specified, and α of 0.05 was used for all tests.

Test characteristics for various DT threshold values were determined using the approach of Jacobsen PB et al. [26]. We defined clinically significant distress as a HADS total score ≥ 15, and, using that reference value, constructed receiver operating characteristics (ROC) curves for DT sensitivity and 1-specificity for each DT value (range 0–10) (Figure 1). We report the area under the curve for our ROC curve, as well as the test characteristics sensitivity and specificity of each DT value for distress.

Statistical calculations were performed using SAS v9.4 (Cary, NC, USA) and our ROC curve was generated using the web-based ROC Plot app [34].

## 3. Results

From September 2020 through May 2021, 211 patients consented to and were screened for participation. Nine lacked a non-hematologic cancer diagnosis; two were not currently receiving or starting chemotherapy within 14 days of enrollment; two were receiving their last planned cycle of chemotherapy on the day of enrollment; one was felt by study staff to be unable to consent to participation; and one participant was too ill to participate after consenting. Of the remaining 196 participants, all completed study procedures at enrollment, 173 completed procedures at second visit, and 150 completed procedures at a third visit. The median time between visits was 22 days (interquartile range: 21–28 days).

Median age for all participants was 52.3 years, and 152 (77.6%) were female (Table 1). Most of the cohort self-identified as Black race (*n* = 157, 80.1%). IsiZulu was the most common primary language (*n* = 56, 28.6%), while English was the primary language of just 38 (19.4%). Participants with breast cancer made up over half of the study cohort (*n* = 102, 52.0%), and those with colorectal cancer were the next most common (*n* = 29, 14.8%). By self-report, 48 participants (24.5%) had comorbid hypertension, and 33 participants (16.8%) had comorbid HIV.

All symptom items showed construct validity by association with the European Organization for Research and Treatment of Cancer Core Quality of Life Questionnaire, version 3.0 (QLQ-C30) score (all *p*-values < 0.0001) (Table 2). Fatigue and nausea were strongly associated (i.e., r ≥ 0.5) with QLQ-C30 score, while fever, cough and diarrhea were weakly associated (i.e., 0.3 > r ≥ 0.1). The remaining symptoms were all moderately associated (i.e., 0.5 > r ≥ 0.3). All symptom items were also significantly associated with all five QLQ-C30 functional sub-scale scores expect diarrhea, which was only significantly associated with emotional and social functioning (Appendix A). The pain, fatigue, dyspnea, nausea, vomiting, constipation and diarrhea PRS-SA items were all strongly associated with their corresponding QLQ-C30 symptom item scales.

All symptoms except mucositis also showed construct validity by association with ECOG performance status (PS) (highest significant *p* = 0.03) (Table 2). All symptoms associated with performance status showed at least a moderate effect size (i.e., Cohen d ≥ 0.5), except diarrhea.

In evaluation for responsiveness, changes in consecutive symptom scores for all items but cough showed association with changes in QLQ-C30 score (highest significant *p*-value = 0.045) (Table 3). The standardized mean scores from all 11 symptom items also showed a significant downward trend across negative, neutral, and positive of the Global Impression of Change (GIC) response groups (*p* = 0.03) (Figure 2).

Of the 175 participants who completed the PRS-SA after receiving at least one cycle of chemotherapy, 110 (62.9%) reported at least one symptom of grade 3–4 severity (Table 4). Grade 3–4 symptoms impacting at least 10% of patients receiving chemotherapy included pain, fatigue, constipation, peripheral neuropathy, oral mucositis and fever.

The receiver operating characteristics (ROC) curve for the modified distress thermometer’s (DT) identification of clinically meaningful anxiety and depression, as measured by the Hospital Anxiety and Depression Scale (HADS), showed an area under the curve (AUC) of 0.76 (Figure 3). DT scores of ≥4 had 82% sensitivity and 55% specificity for depression and anxiety. As a sensitivity analysis, we also tested DT test characteristics for the individual HADS sub-scores for both anxiety and depression, using a sub-score of ≥8 as signifying meaningful symptoms in both cases. In this post hoc analysis, DT scores of ≥4 had a 78% sensitivity and 53% specificity for anxiety and an 82% sensitivity and 52% specificity for depression. Of the 196 participants, 151 (77.0%) reported a DT score of ≥4 on at least one instance of the PRS-SA. Among those reporting a DT score of 4 or higher, the most commonly selected problems contributing to suffering were finances (*n* = 90, 59.6%), stress (*n* = 88, 58.3%) and transportation (*n* = 83, 55.0%).

Of the 167 participants who preferred to complete the PRS-SA and other study instruments in English, 153 (91.6%), 152 (91.0%) and 147 (88.0%) reported understanding, reading and writing English well or very well, respectively. However, during the 458 times the PRS-SA was administered in English, the study interviewer reported needing to read half or more of the instrument to the participant in 210 (45.9%) cases. Similarly, among the 29 participants choosing to complete study instruments in isiZulu, 29 (100%), 26 (89.7%) and 26 (89.7%) reported understanding, reading and writing isiZulu well or very well, respectively. For this group, study interviewers reported reading half or more of the PRS-SA in 41 (73.8%) of the 61 instances it was completed. In a post hoc sensitivity analysis, neither the distribution of DT scores or of reported severity of any symptom item significantly differed between participants who did and did not require any assistance with reading the PRS-SA.

## 4. Discussion

This study is the first in South Africa to validate a locally relevant PROM for patients with solid tumors. We demonstrate construct validity for all symptom items, and all items showed at least a medium-sized association with either QLQ-C30 summary score or ECOG PS, except for diarrhea which showed a small-sized association with both. All individual symptom items except cough also showed responsiveness to change in health-related quality of life (HRQOL). The full set of items was responsive to changes in patients’ GIC score. Our approach to establishing construct validity and responsiveness of our symptom items was modeled after that of the PRO-CTCAE Study Group [18]. That group also found medium to strong associations and responsiveness to HRQOL with most of their PROM symptom items in a United States-based cohort. Interestingly, loose stool showed only weak effect sizes in their study as well.

Responses to the modified DT were also remarkably consistent with other populations worldwide. Using an international standard to detect clinical depression or anxiety, our ROC curve for the DT has an AUC of 0.76. Reported AUCs from DT validation studies in 33 other countries range from 0.63 to 0.88 [15]. Using a cut-off DT value of ≥4 to identify depression/anxiety, our measured sensitivity of 82% is also consistent with international experience, though our specificity of 55% is lower than typically reported. During the process of drafting the PRS-SA, a group of local oncology providers advised that the term “distress” would not be universally understood and should be changed to “suffering” for the modified DT. This change may contribute to the item’s reduced specificity for predicting depression/anxiety. In practice, it may be appropriate to ask several additional screening questions for emotional distress prior to referring patients with a DT score ≥4 for psychologic or stress management interventions.

While not the primary interest of this study, it is notable that >60% of participants reported at least one grade 3 or higher chemotherapy-related symptom, and >75% of participants reported DT scores ≥4. Chemotherapy adverse event (AE) rates in sub-Saharan African populations are rarely published, but our findings are consistent with the >70% rate of grade 3–5 events at one hospital in Ethiopia and much higher than the ~30% grade 3–4 event rate among breast cancer patients treated in Senegal [35,36]. Self-reported grade 3–4 AEs rates are comparable in high-income countries as well, with two studies from United States and one from Australia reporting rates at 63%, 51% and 62%, respectively [7,37,38]. Over half of those reporting significant distress cited financial concerns as a major stressor. Unfortunately, this is also consistent with other African populations. Studies in Uganda, Nigeria, Eswatini and South Africa have all previously identified financial strain as major contributors to distress, non-adherence with cancer related care or both [39,40,41,42,43,44].

Following validation, it will be necessary to pilot and study implementation of general use of the PRA-SA at our clinic and similar settings. In high-income environments, provider-level barriers to PROM implementation have included skepticism regarding their value or necessity, underdeveloped strategies for interpreting or reacting to symptoms and distress, insufficient time or staffing to review reports, underuse of information technology solutions for presenting and tracking response over time and fear that PROMs depersonalize doctor–patient interactions [45,46,47,48]. Resource-constrained settings are potentially vulnerable to all these issues. Task-shifting strategies and standardized clinical pathways for responding to specific symptoms may be useful for overcoming staff and resource shortages and building capacity to appropriately respond to patients’ reports of treatment-related toxicity.

Our results point to another major potential barrier to implementation of PROM use in general clinical care in South African oncology clinics: literacy. Despite ~90% of participants reporting they could read and write well or very well in either English or isiZulu, 46% and 74% required assistance reading at least half of the English and isiZulu PRS-SA, respectively. Formal literacy evaluation in South African patients has shown wide discrepancies between the ability to read or pronounce English medical terms and the ability to understand their meaning [49]. To reduce the need for assistance completing the PRS-SA, future versions might employ pictograms, as with Global Oncology’s cancer education materials, or audio recordings administered via tablet computers [50]. These sorts of solutions might be even more essential in rural areas, where literacy rates are lower than our urban clinic [51]. We wanted to ensure multiple language options for participation in this validation study. While we were not adequately powered to separately validate the English and isiZulu versions of the PRS-SA, exploratory analyses of only the English version showed the similar results for construct validity or reliability as analysis of the full cohort.

Given the myriad of other challenges resource-constrained healthcare systems face in providing effective, high-quality cancer care, it will also be necessary to demonstrate that PROM use for monitoring chemotherapy toxicity maintains the ability to improve patient quality of life, treatment adherence and clinical outcomes. As that work proceeds in South Africa, we offer our validated tool here in hope that researchers and clinicians working in similar settings will take up its use and simultaneous study.

## 5. Conclusions

In conclusion, our novel PROM, the PRS-SA, demonstrated construct validity and responsiveness in the measurement of chemotherapy-related symptom severity and typical sensitivity to clinically meaningful distress among solid tumor patients receiving chemotherapy at a public oncology clinic in Johannesburg, South Africa. Patients reported both a high burden of grade 3–4 symptoms and distress. Incorporating the PRS-SA into clinical use in South Africa may provide a low-cost, low-tech way to improve cancer patient QoL. Based on these initial findings, the instrument’s implementation, and impact merit further study. We anticipate that low patient literacy will present a major barrier to the PRS-SA’s routine use; creative solutions will be needed to increase accessibility.

## Figures and Tables

**Figure 1 cancers-14-00095-f001:**
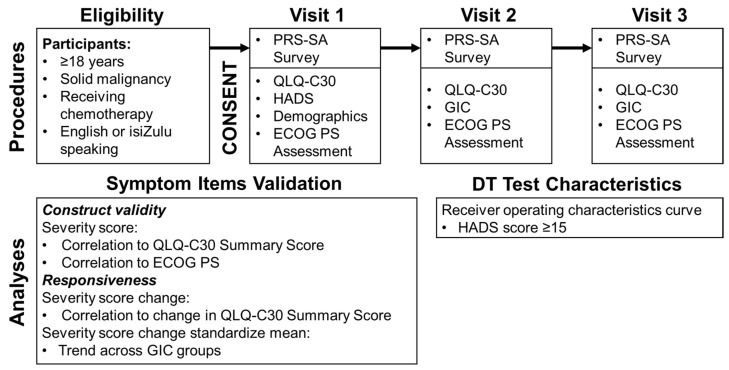
Procedures and analyses used for PRS-SA validation. (Abbreviations: ECOG PS-Eastern Cooperative Oncology Group performance status, GIC-Global Impression of Change questionnaire, HADS-Hospital Anxiety and Depression Scale, PRS-SA-Patient Reported Symptoms-South Africa, QLQ-C30-European Organisation for Research and Treatment of Cancer Core Quality of Life Questionnaire, version 3.0).

**Figure 2 cancers-14-00095-f002:**
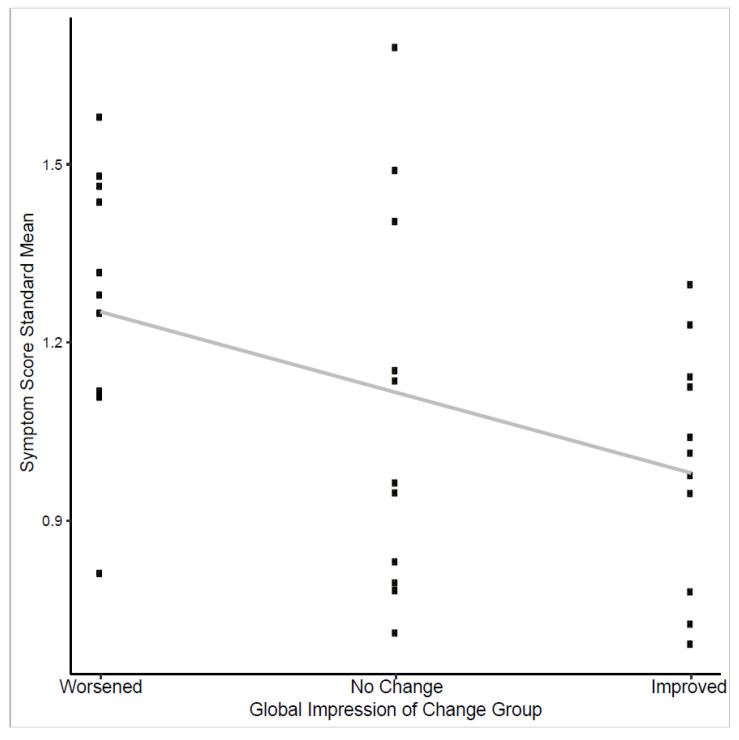
Trend test of symptom score standardized means and global impression of change response.

**Figure 3 cancers-14-00095-f003:**
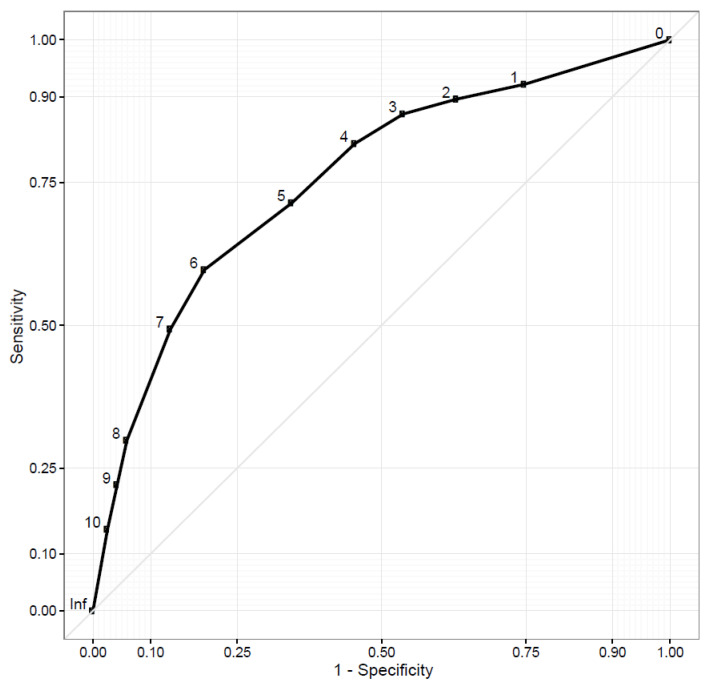
Receiver operator characteristics curve for distress thermometer detecting depression/anxiety.

**Table 1 cancers-14-00095-t001:** Patient demographics and clinical characteristics. (Abbreviations: COPD-Chronic obstructive pulmonary disease, IQR-Interquartile range, PRS-SA-Patient Reported Symptoms-South Africa).

Characteristic	Full Cohort *n* = 196
	*n* (%)
**Sex**	
Female	152 (77.6)
Male	44 (22.5)
**Age in years (Median, IQR)**	52.3 (43.3–61.3)
**Race**	
Black	157 (80.1)
White	19 (9.7)
Mixed Race	17 (8.7)
Asian	2 (1.0)
Other	1 (0.5)
**Primary/Home Language**	
IsiZulu	56 (28.6)
English	38 (19.4)
Sesotho	30 (15.4)
Sepedi	10 (5.1)
Tsonga	10 (5.1)
Tswana	10 (5.1)
Other	41 (20.9)
Declined to Answer	1 (0.5)
**PRS-SA Language Preference**	
English	167 (85.2)
IsiZulu	29 (14.8)
**Response to “How Well Do You Read English?” Question** (*n* = 167)	
Very Well	87 (52.1)
Well	65 (38.9)
Not Well	14 (8.4)
Not At All	1 (0.6)
**Response to “How Well Do You Read isiZulu?” Question** (*n* = 29)	
Very Well	13 (44.8)
Well	13 (44.8)
Not Well	3 (10.3)
Not At All	0 (0.0)
**Relationship Status**	
Single (Never married)	81 (41.3)
Married/Partnered	72 (36.7)
Divorced/Separated	20 (10.2)
Widowed	23 (11.7)
**Employment Status**	
Unemployed	97 (49.5)
Employed	56 (28.6)
Retired	41 (20.9)
Student	2 (1.0)
**Cancer Site**	
Breast	102 (52.0)
Colorectal	29 (14.8)
Kaposi and Other Sarcomas	11 (5.6)
Stomach	10 (5.1)
Prostate	8 (4.1)
Cervical	7 (3.6)
Lung	7 (3.6)
Ovarian	5 (2.6)
Pancreatic and Biliary	6 (3.1)
Liver	4 (2.0)
Head and Neck	2 (1.0)
Skin/Melanoma	2 (1.0)
Esophagus	1 (0.5)
Uterine	1 (0.5)
Vulvar	1 (0.5)
**Cancer Treatments Received Prior to Enrollment**	
Chemotherapy	41 (20.9)
Surgery	66 (33.7)
Radiation	29 (14.8)
**Comorbidities (Self-Report)**	
Hypertension	48 (24.5)
HIV	33 (16.8)
Diabetes	13 (6.6)
Asthma/COPD	2 (1.0)
Another Cancer	1 (0.5)

**Table 2 cancers-14-00095-t002:** Symptom item severity associations with QLQ-C30 scores and ECOG performance status (construct validity). (Abbreviations: ECOG PS-Eastern Cooperative Oncology Group performance status, QLQ-C30-European Organisation for Research and Treatment of Cancer Core Quality of Life Questionnaire, version 3.0).

Anchor Item				
Symptom	QLQ-C30 Summary Score		ECOG PS (0–1 vs. 2–4)	
	r	*p*-value	Cohen’s d	*p*-value
Pain	0.46	<0.0001	0.81	<0.0001
Fatigue	0.65	<0.0001	1.17	<0.0001
Fever	0.23	<0.0001	0.51	0.02
Dyspnea	0.45	<0.0001	1.36	<0.0001
Cough	0.24	<0.0001	0.62	<0.0001
Oral Mucositis	0.40	<0.0001	-	0.16
Nausea	0.56	<0.0001	0.80	<0.0001
Vomiting	0.40	<0.0001	1.02	<0.0001
Diarrhea	0.17	<0.0001	0.30	0.03
Constipation	0.40	<0.0001	0.50	0.002
Peripheral Neuropathy	0.41	<0.0001	0.60	0.003

**Table 3 cancers-14-00095-t003:** Association between symptom change and QLQ-C30 change (responsiveness). (Abbreviations: QLQ-C30-European Organisation for Research and Treatment of Cancer Core Quality of Life Questionnaire, version 3.0).

Symptom	*p*-Value
Pain	<0.0001
Fatigue	<0.0001
Fever	0.002
Dyspnea	0.045
Cough	0.4
Oral Mucositis	0.0009
Nausea	<0.0001
Vomiting	<0.0001
Diarrhea	<0.0001
Constipation	0.007
Peripheral Neuropathy	0.002

**Table 4 cancers-14-00095-t004:** Symptoms reported by participants receiving chemotherapy.

Symptom (*n* = 175)	Any (*n*, %)		Grade 3–4 (*n*, %)	
Any	173	98.9%	110	62.9%
Pain	149	85.1%	68	38.9%
Fatigue	156	89.1%	61	34.9%
Constipation	96	54.9%	33	18.9%
Peripheral Neuropathy	116	66.3%	29	16.6%
Oral Mucositis	88	50.3%	24	13.7%
Fever	72	41.1%	18	10.3%
Dyspnea	59	33.7%	16	9.1%
Vomiting	77	44.0%	13	7.4%
Nausea/Anorexia	121	69.1%	12	6.9%
Diarrhea	67	38.3%	6	3.4%
Cough	67	38.3%	5	2.9%

## Data Availability

De-identified data presented in this study are available on request from the corresponding author. The data are not publicly available due to compliance with national privacy laws.

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
