# Peer review of "Validating an Instrument for Direct Patient Reporting of Distress and Chemotherapy-Related Toxicity among South African Cancer Patients"

_cancers, 2021, doi:10.3390/cancers14010095_

Round 1

Reviewer 1 Report

Thank you for the opportunity to review this paper, which describes the validation of a distress and symptom screening tool for South African cancer patients. This is an important initiative for which the authors are to be commended. My feedback is as follows:

  1. Abstract, line 2: "No similar PROMs exist" - do you mean no PROMs exist? Otherwise please elaborate on the PROMs to which these are similar.
  2. Abstract: the findings in relation to literacy and need for assistance completing the PROMs are important and should be mentioned
  3. Introduction or methods/settings and participants: more information about the demographics of the patient population here would be informative, especially with respect to languages spoken and literacy levels for completion of PROMs.
  4. Methods, paragraph 1: It is not necessary to name the authors here. However greater information about the focus group process and its findings are recommended, unless this work is published elsewhere (in which case please include the relevant reference here).
  5. Methods section 2.5: Please elaborate on why correlations were sought between symptoms and the global QLQ C30 summary score, rather than subscales or relevant individual symptom items (table 2)
  6. Table 1: please differentiate subheadings from the items in each category. 
  7. Table 1: include data about the literacy levels of participants
  8. Discussion: the lower specificity of the SA version of the instrument compared to others internationally is an important finding. Please elaborate on the clinical implications of this finding if it were to become used in routine practice
  9.  Discussion: the authors comment that the study was not adequately powered to validate the instrument in both English and isiZulu. Was there adequate power in either language? Please elaborate on what this means for next steps in the development of the instrument.
  10. Discussion: there is relatively brief comment made on capacity of services to respond to issues raised by patients in their PROMs. I suspect this will prove even greater than in high income nations, and warrants further comment here.

Reviewer 2 Report

Thank you for the opportunity to review this manuscript.
Overall, it is well written. I it appears novel since there is no such instrument in Africa.

I have a few minor suggestions or comments.

  1. I did not find the instrument. The instrument should in be included as a supplement and included a rein the main text. 

Reviewer 3 Report

The paper is interesting and presents a significant advancement with respect to the state-of-the art. Given the national domain of the instrument validation, it is of particular relevance for South African population, although limited for the countries other than that, as mostly happens with validation studies.

However, the paper is well presented, and pleasant to read. I would not suggest any particular improvement, if not a more in depth insight into the methodological part concerning the validation, possibly helpful to replicate the study. Eventual figures/block diagrams could be useful to better explain the whole validation process.

After this integration, I would recommend the acceptance of the manuscript.
